

# Parent Hamiltonian reconstruction
# of Jastrow-Gutzwiller wavefunctions

**Xhek Turkeshi**[1,2,3⋆] **and Marcello Dalmonte**[1,2]

**1** The Abdus Salam International Centre for Theoretical Physics,
Strada Costiera 11, 34151 Trieste, Italy
**2** SISSA, via Bonomea 265, 34136 Trieste, Italy
**3** INFN, sezione di Trieste, 34136 Trieste, Italy

⋆ xturkesh@sissa.it

## Abstract

Variational wave functions have been a successful tool to investigate the properties of quantum spin liquids. Finding their parent Hamiltonians is of primary interest for the experimental realization of these strongly correlated phases, and for gathering additional insights on their stability. In this work, we systematically reconstruct approximate spin-chain parent Hamiltonians for Jastrow-Gutzwiller wave functions, which share several features with quantum spin liquid wave functions in two dimensions. Firstly, we determine the different phases encoded in the parameter space through their correlation functions and entanglement properties. Secondly, we apply a recently proposed entanglement-guided method to reconstruct parent Hamiltonians to these states, which constrains the search to operators describing relativistic low-energy field theories - as expected for deconfined phases of gauge theories relevant to quantum spin liquids. The quality of the results is discussed using different quantities and comparing to exactly known parent Hamiltonians at specific points in parameter space. Our findings provide guiding principles for experimental Hamiltonian engineering of this class of states.



# 1   Introduction

Variational wave functions play a key role in the understanding of quantum phases of matter [1–8]. A paradigmatic example is Laughlin wave functions [5], which can be formulated as parametric Jastrow states reproducing several key features of certain fractional quantum Hall effects [9]. Shortly after this, resonating valence bond (RVB) states have been employed as effective descriptions of high-temperature superconductors [6, 7, 10], and later on, have been linked to fractional quantum Hall physics in Ref. [8]. These early successes boosted variational wave functions as theoretical tools to provide simple pictures for a variety of quantum phases, including topological matter, low-dimensional systems, and tensor networks [11–15].

Perhaps, among these applications, one of the most fruitful has been in the field of quantum spin liquids [16–22]. These are quantum phases characterized by strong correlations and long-range entanglement among arbitrary far subregions of the system [23], and for these reasons, semi-classical pictures fail in describing the phenomena involved. Variational wave functions have been used to distill generic properties such as correlation functions and entanglement [14].

Interestingly, despite the conceptual simplicity of Jastrow wave functions, it is often challenging to find the corresponding parent Hamiltonians - that is, the Hamiltonians supporting these wave functions as ground states. The major obstruction is that, given a Hamiltonian on a lattice (possibly with frustration terms), quantum fluctuations may cooperate and induce an ordered ground state. This phenomenon is typically referred to as "order-by-disorder" [13]. This problem is of primary importance also due to the latest experimental breakthrough in quantum engineering of synthetic systems [24–28]. In fact, the high degree of interaction tunability of these platforms offers new perspectives and possibilities in otherwise hardly achievable phases of matter, including spin liquids, once parent Hamiltonians are (approximately) identified.

Most of the works in parent Hamiltonian construction studied specific variational states using insightful analytic manipulations [29–44]. Very recently, a series of novel techniques based on systematic approaches have been considered in Ref. [45–50]. Indeed, the authors of the latter works introduced new efficient computational algorithms, which remarkably scale polynomially in the system size when restricting the search to *local* Hamiltonians that have a given initial state as the input eigenstate. To benchmark their techniques, they considered the ground state of some *a priori* known Hamiltonian as input and checked if the output

reconstructed operator coincided with that Hamiltonian. So far, however, there have been no applications of such methods to generic spin liquid variational wave functions, whose parent Hamiltonians are still undetermined.

The present work is the first step in this direction. For concreteness, here we study the class of 1D Jastrow-Gutzwiller variational wave functions [30,51]. These states share two key features with their two-dimensional cousins employed as effective descriptions of quantum spin liquids: they describe extensive superpositions over some (spatially local) state basis, and they have in general as weights analytic functions of the space coordinates. Despite their common appearance, their parent Hamiltonians are not known except for a few fine-tuned cases, amenable to exact solutions. We use an entanglement-guided algorithm presented in Ref. [50] to search local parent Hamiltonians for these states. This method relies on the Bisognano Wichmann theorem [52,53], a quantum field theory result that links systematically the local Hamiltonian density to its ground state reduced density matrix. Its advantage with respect to the other above-mentioned techniques resides in certifying the input state as the ground state of the reconstructed parent Hamiltonian. Indeed, although the methods in Ref. [45–49] are of broader applicability (for instance, they allow for extensions to time-dependent problems), they typically certify the ansatz state to be a generic *eigenstate*, and not the *ground state*, of the output operator. The main disadvantage is that the method is not applicable in case the wave function cannot be cast as the ground state of Hamiltonian operator supporting low-energy relativistic excitations.

Since the Bisognano-Wichmann technique requires the input state to exhibits relativistic low lying physics, we first investigate the entanglement and correlation properties of these wave functions, identifying a region where the algorithm is expected to perform better. In this regime, we obtain local approximate parent Hamiltonian searching through different algebras of local operators. To check our results, we computed the relative entropy, the correlation functions and the overlap between their ground state and the Jastrow-Gutzwiller wave functions, obtaining fidelities ranging between 95% to over 99%. In addition, we computed the relative error between the ground state energy and the Jastrow-Gutzwiller variational energy of the reconstructed Hamiltonian. In all the considered cases, the relative error is less than 1%, even in the extrapolated thermodynamic limit. We perform systematic searches by increasing both system sizes and interaction range. These results suggest that the exact, yet unknown, parent Hamiltonians of these states exhibit long-range features.

In addition, the method allows us to perform direct parent Hamiltonian searches utilizing simple long-range interactions in the form of monotonous power-law potentials. We find that, while considerably improving the parent Hamiltonian search, such simple long-range interactions are not always sufficiently rich to capture the (unapparent) complexity of Jastrow-Gutzwiller wave functions. These results indicate that the search for exact - albeit long-ranged - parent Hamiltonians for 2D Jastrow-Gutzwiller might be particularly challenging, a fact which is compatible with the scarcity of exact results in this context (with some notable exceptions, see Ref. [29,40]).

The remaining of this paper is structured as follows. In Section 2 we introduce the Jastrow-Gutzwiller states and discuss their physical content through participation spectrum, entanglement entropy and correlation functions. In Section 3 we summarize the Bisognano-Wichmann Ansatz method which we employ in Section 4 to reconstruct various parent Hamiltonians for the above-considered states. The last section is for conclusions and outlooks.

## 2 Jastrow-Gutzwiller wave functions

### 2.1 Model wave functions

The Jastrow-Gutzwiller (JG) wave functions are paradigmatic states appearing in several contexts, from integrability to topology (e.g. Laughlin states), to quantum spin liquids. They are characterized by an extensive superposition of spatially local states, and the local weights of the wave functions are captured by polynomials. Throughout this paper, we investigate the one-dimensional case defined on a periodic chain $\Lambda$ of length $L$. This setting permits the understanding of finite-volume effects in a systematic manner, as well as enables comparison to exact results.

Let us introduce the wave functions of interest, through the variables $n_i \in \{0, 1\}$ defined at each site $i \in \Lambda$. In the basis $\{|n_1 n_2 \ldots n_L\rangle\}$, these states read:

$$|\Psi_\alpha\rangle = \sum_{\mathcal{P}_N\{n\}} \psi_\alpha(\{n\}) |n_1 n_2 \ldots n_L\rangle, \tag{1}$$

$$\psi_\alpha(\{n\}) = \frac{1}{Z} (-1)^{\sum_{i=1}^L i n_i} \prod_{1 \le i < j \le L} \sin\left(\frac{\pi}{L}(j-i)\right)^{\alpha n_i n_j}.$$

Here the sum is over combinations $\mathcal{P}_N\{n\}$ constrained by $\sum_i n_i = N$. Pictorially, the $\{n_i\}$ variables are occupation numbers of hard-core bosons living on the lattice. The real parameter $\alpha$ and the filling fraction $\nu = N/L$ control the properties of the states. For specific combined values of $\nu$ and $\alpha$, conformal field theory calculations have been used to derive exact results pertaining the parent Hamiltonians of these states [42–44, 54, 55]. Throughout this paper, we will consider exclusively the half-filling case $\nu = 1/2$ and $L$ even; the main motivation being that, in spin language, this regime captures both paramagnetic and antiferromagnetic phenomenology.

Within this setting, exact results are available only for $\alpha \in \{0, 1, 2\}$. In Ref. [56], it was proven that $\alpha = 0$ corresponds to the XXZ chain at $\Delta = -1$, while the state at $\alpha = 2$ is the ground state of the Haldane-Shastry Hamiltonian [30, 31]. The case $\alpha = 1$ corresponds to a (symmetrized) Slater determinant, and its parent Hamiltonian is a free fermionic one (up to boundary contributions).

### 2.2 Participation spectrum

To obtain insights for generic values of $\alpha$, it is instructive to rephrase Eq. (1) in the language of participation spectroscopy [58–62]. This consists of rewriting the wave functions Eq. (1) in a pseudo-energy fashion:

$$\psi_\alpha(\{n\}) = \langle n_1 n_2 \ldots n_L | \Psi_\alpha \rangle \tag{2}$$

$$\equiv \frac{e^{-H_\alpha[\{n\}]}}{Z_\alpha}. \tag{3}$$

In the last equality, we defined the function $H_\alpha[\{n\}]$:

$$H_\alpha[\{n\}] = \alpha \sum_{1 \le i < j \le L} n_i n_j V(i,j) + E_0[\{n\}], \tag{4}$$

$$V(i,j) = -\log\left[\sin\left(\frac{\pi}{L}(j-i)\right)\right], \tag{5}$$

$$E_0[\{n\}] = \log\cos\left(\sum \pi i n_i\right). \tag{6}$$

The functional coefficient $E_0[\{n\}]$ is an energy constant, while $V(i,j)$ is a logarithmic interaction between occupied particles mediated by chord distances. Thus, we recognize $H_\alpha[\{n\}]$ to be a 2D Coulomb gas (classical) Hamiltonian constrained in a 1D circular lattice [54, 57]. Analogously, the wave function normalization $Z_\alpha$ is a classical partition function:

$$Z_\alpha^2 = \sum_{\mathcal{P}_N(\{n\})} e^{-2H_\alpha[\{n\}]}. \tag{7}$$

The parameter $\alpha$ plays the role of an inverse temperature and controls the leading weights in the JG states. The modulus squared coefficients in Eq. (1):

$$p_\alpha(\{n\}) \equiv |\psi_\alpha(\{n\})|^2 = \frac{e^{-2H_\alpha[\{n\}]}}{Z_\alpha^2}, \tag{8}$$

are Boltzmann weights with classical Hamiltonian $2H_\alpha$ and partition function $Z_\alpha^2$. The pseudo-energies of $2H_\alpha$ are collectively named participation spectrum and denoted $\varepsilon(\{n\})$.

The ground state $\varepsilon_{\min}$ determines the larger weights in the sum Eq. (1). For $\alpha > 0$, the Hamiltonian favors repulsion among particles, constrained by the half-filling condition. Thus, the most probable configurations come from alternating occupation numbers. At negative inverse temperature $\alpha < 0$ the dominant coefficients are those maximizing the number of occupied nearest neighboring sites. For both cases, such configurations are not unique but degenerate, and for large values of $\alpha$ these states are expected to be the most relevant contributions to the Jastrow-Gutzwiller wave functions. Consequently the JG state are captured by the coherent superposition of these degenerate configurations, which leads, for $\alpha \gg 1$ and $\alpha \ll -1$, respectively to antiferromagnetic and ferromagnetic Greenberger–Horne–Zeilinger (GHZ) states [63]:

$$|\Psi_\alpha\rangle_{\alpha \gg 1} \simeq \frac{1}{\sqrt{2}} (|1010\ldots10\rangle + |0101\ldots01\rangle),$$

$$|\Psi_\alpha\rangle_{\alpha \ll -1} \simeq \frac{1}{\sqrt{L}} \sum_{i=1}^{L} |\ldots 0_{i-1} 1_i 1_{i+1} \ldots 1_{i+L/2} 0_{i+L/2+1} \ldots\rangle. \tag{9}$$

The former state is usually dubbed Néel/anti-Néel state and corresponds to a global Schrödinger cat state. Apart from these extreme limit, at intermediate values of $\alpha$ the system exhibits competing weights, which render rigorous analytical arguments demanding.

To test this heuristic argument, we consider the gap $G = \varepsilon_{\min} - \varepsilon_{1^{\text{st}}}$ between the ground state energy of Eq. (4) and its first excited energy, which we refer to as participation gap. Let us discuss the case $\alpha > 0$. It is convenient to introduce the number of ferromagnetic domain walls as the number of consecutive occupied/unoccupied sites $N_{\text{dws}}$. For example $N_{\text{dws}}(|010101\rangle) = 0$, while $N_{\text{dws}}(|011001\rangle) = 2$. The Néel and anti-Néel states, i.e. the most probable states, are the only ones with $N_{\text{dws}} = 0$, and all other pseudo-energy excitations can be easily labelled with this number. In Fig. 1 we present the participation spectrum of the JG states for $\alpha = 2, 6$ and $L = 16$. The gap $G$ between the most probable and the second most probable state increases linearly with $\alpha$, with an exactly computable $L$-dependent constant $g_L$. This saturates a thermodynamic value[1] $g_\infty$ already for modest system sizes.

It is important to emphasize one aspect that is relevant in determining the system properties in the thermodynamic limit. The ground state pseudo-energy with alternating occupied

---

[1] We get an analytic expression for the constant:

$$g_\infty = 2 \lim_{L \to \infty} \log\left(\frac{\sin(2\pi/L) \prod_{r=1}^{L/2-2} \sin(2r\pi/L)}{\sin(\pi/L) \prod_{r=1}^{L/2-2} \sin((2r+1)\pi/L)}\right) \simeq 0.9031654195\ldots, \tag{10}$$

where the ellipsis indicate further computable digits.

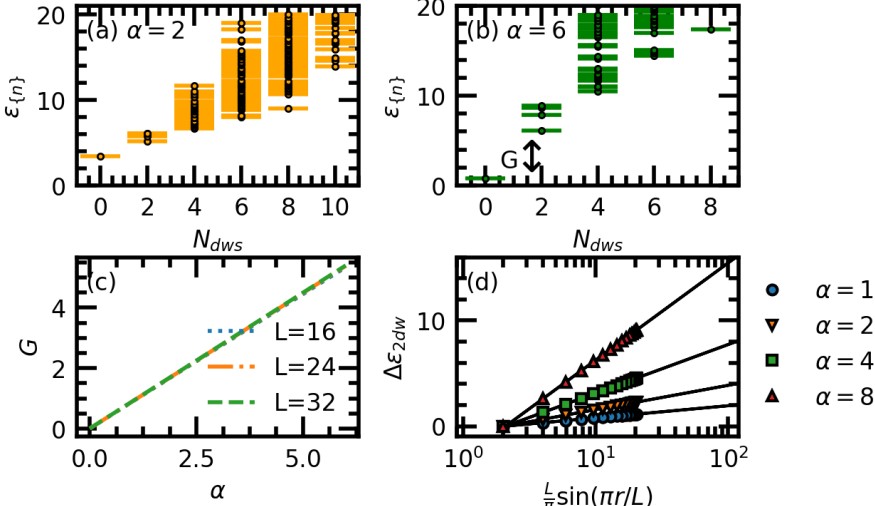

Figure 1: (a,b) The participation spectrum $\epsilon(\{n\})$ for $\alpha = 2, 6$ and $L = 16$. The spectrum is indexed using the number of domain walls configurations $N_{dws}$. (c) The participation gap $G$ increases linearly with $\alpha$, with a coefficient that saturates to a constant $g_\infty$ already at modest system sizes. (d) Pseudo-energy differences between two domain walls as a function of the domain-wall separation $r$. This is a measure of the confining potential between domain walls. The black solid line is the Luttinger liquid prediction [60] with Luttinger parameter $K = 1/\alpha$. The fit describe extremely well our data, for $4 \leq L \leq 64$.

sites is doubly degenerate for every system size. Instead, although the configurations with domain walls are exponentially suppressed in $\alpha$, their degeneracy scales linearly with system size. In particular at $L \sim \exp(c\alpha)$ for some constant $c$, we expect a competing and non-trivial behavior between the Néel sector and the first excited sector. This has potentially relevant consequences, which are difficult to predict within the present study. In particular is unclear what kind of effects this pseudo-energy thermodynamics can induce on quantum observables such as, e.g., correlation functions.

At a practical level, our results are consistent with the intuition above, that the Néel state predominately contributes for large $\alpha$. In order clearly see the effects of the aforementioned thermodynamic competition for $\alpha = 6$, we would have needed around $L \sim 10^4$ sites. The large gap for any computable finite $L$ considered, renders these excited state sectors negligible.

The results for $\alpha < 0$ are analogous to the latter, whereas the most probable configurations are the ferromagnetic ones and the excited pseudo-energy states are obtained as functions of antiferromagnetic domain walls, i.e. number of alternating occupied/unoccupied sites. However the most probable states there are $L$-degenerate: in the thermodynamics of the Coulomb gas this implies the low-lying pseudo-energy excitation are negligible even at small negative values of $\alpha$.

Finally, from the substructure of the $N_{dws} = 2$ sector we can extract how these domain walls interact. In particular, the pseudo-energy difference $\Delta\varepsilon_{2dw} = \varepsilon_{2dw}(r) - \varepsilon_{2dw}(2)$ between domains separated by a distance $r$ and those close together ($r = 2$) has been used for local antiferromagnetic quantum Hamiltonian systems to distinguish between critical and symmetry broken phases of matter. In the former case, the domain walls are logarithmically confined with the separation distance; instead, in the latter this confining is linear. Moreover, the prefactor of this potential for 1D Luttinger liquids [66, 67] is related to the Luttinger parameter. This has been tested in Ref. [60], where its authors analyze the XXZ chain.

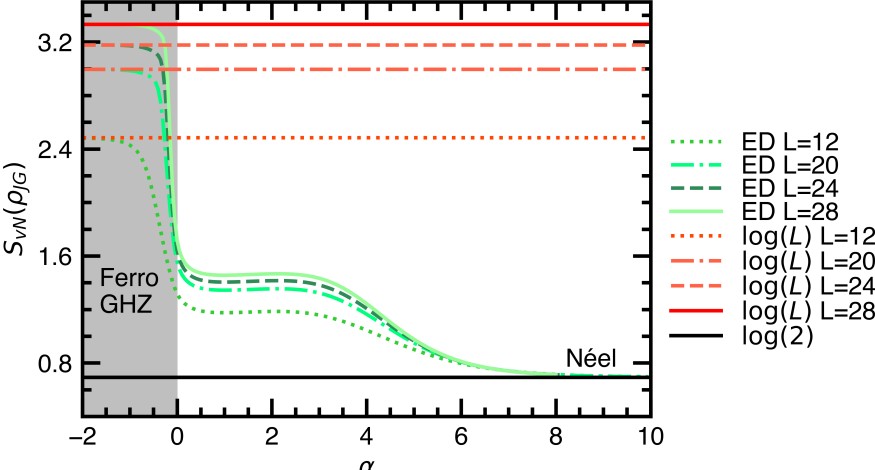

Figure 2: We plot the entanglement entropy $S_{\text{vN}}(\rho_{JG})$ at different values of $\alpha$ for $L = 12, 20, 24, 28$. Here $\rho_{JG}$ is the half-system reduced density matrix of the JG state. The results in green line are obtained through ED using symmetry restrictions. The red lines are the ferromagnetic GHZ predictions for the corresponding system sizes, while the black one is the Néel/anti-Néel cat state entanglement entropy.

Because of the explicit form of the classical Hamiltonian density Eq. (6), the interaction between two domain walls is expected to be logarithmic with their separation distance (Fig. 1, panel (d)). By analogy with the XXZ phenomenology, one is tempted to conclude the JG states are 'critical'. If furthermore one assumes these states are representatives of Luttinger liquids, the fitted pre-factor suggests a Luttinger parameter $K = 1/\alpha$. The latter statement has been recently conjectured [64]. This hypothesis is supported by CFT arguments [56] and from studies on the Resta polarization [65]. Here the authors estimate $\alpha_c = 4$ as critical value separating a conducting Luttinger phase to an insulating Néel ordered phase. Our data do not exhibits any transition point in the participation gap, nor a clear distinction between a gapped and a gapless phase. As remarked earlier, this may be due to a finite size effect, which we are not able to resolve at computationally affordable system sizes. In fact it is possible that the $N_{\text{dws}} = 2$ domain walls sector results as decoupled for physical observables of the system, after a critical value of $\alpha$. At present, however, the consequences of the participation spectroscopy to physical observables are unclear, and further studies are needed in this direction. In the next two subsections we improve our understanding of the Jastrow-Gutzwiller wave functions by numerically studying the entanglement entropy and the correlation functions. We focus on these properties among others because they serve in the reconstruction technique and its quality checks. The considered system sizes suggests the existence of a critical phase between a Néel and a ferromagnetic GHZ regimes. Using finite size scaling we can bound the former in the interval $\alpha \in (0, 4.3)$.

## 2.3 Entanglement entropy

In this subsection, we discuss the entanglement entropy properties of the JG states (for related studies of Rényi entropies in 2D, see Ref. [14]). Entanglement is a fundamental quantity measuring quantum correlations among subregions of the system [68–73]. For pure states, this is determined by the spectrum of the reduced density matrix [74, 75]. This operator is defined by giving a bipartition of the chain $\Lambda = A \cup \bar{A}$ and a state $|\Phi\rangle$:

$$\rho_A = \text{tr}_{\bar{A}} |\Phi\rangle \langle \Phi| . \tag{11}$$

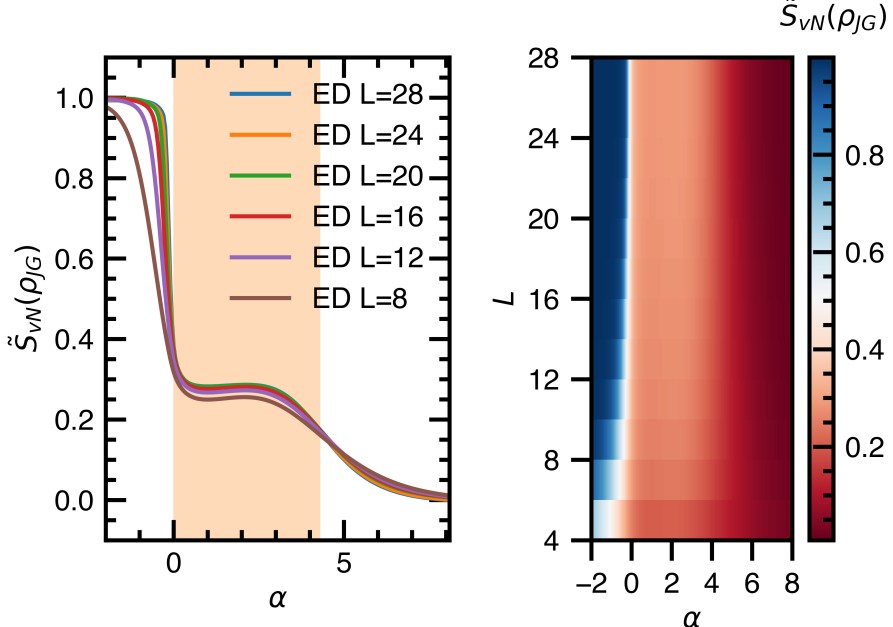

Figure 3: (Left) The function $\tilde{S}_{\mathrm{vN}}(\rho_{JG})$ is plotted versus the parameter $\alpha$ for different $L$. Here $\rho_{JG}$ is the half-system reduced density matrix of the JG state. The shaded area corresponds to states in the critical regime. (Right) Entanglement entropy of the JG reduced density matrix. The critical region extends for $\alpha \in (0, 4.30)$.

Given its spectrum $\sigma(\rho_A)$, we define the von Neumann entropy by:

$$S_{\mathrm{vN}}(\rho_A) = -\mathrm{tr}_A \rho_A \log \rho_A = - \sum_{\lambda \in \sigma(\rho_A)} \lambda \log \lambda. \tag{12}$$

This function is a *bona fide* measure of entanglement for pure states when the Hilbert space factorizes in a tensor product form, $\mathcal{H} = \mathcal{H}_A \otimes \mathcal{H}_{\bar{A}}$, and for this reason is usually referred to as entanglement entropy [76, 77]. Fixing $A = \{1, 2, \ldots, L/2\}$, we compute through exact diagonalization (ED) the von Neumann entropy for the state Eq. (1). We check the GHZ limits by comparing with the analytic calculations for the states in Eq. (9):

$$S_{\alpha<0}(\rho_A) \simeq \log L, \quad S_{\alpha \gg 1}(\rho_A) \simeq \log 2. \tag{13}$$

The agreement is shown in Fig. 2. We isolate an intermediate region between the GHZ regimes by introducing the function:

$$\tilde{S}_{\mathrm{vN}} \equiv \frac{S_A(\rho_{JG}) - \log 2}{\log L - \log 2}. \tag{14}$$

We plot this function in Fig. 3. Within this interval, $\tilde{S}_{\mathrm{vN}}$ is logarithmic, with a pre-factor close to $1/3$. This is consistent with exact solutions, where the systems display a critical regime. For instance, at $\alpha = 1$ the system is a linear combination of Slater determinant. At this point the JG state correspond to a free fermion gas and the entanglement entropy can be computed analytically [78, 79]:

$$S_{\alpha=1} = \frac{c}{3} \log\left(\frac{L}{2}\right) + o(1). \tag{15}$$

Here $c$ is the central charge ($c = 1$ for free fermions) and the sub-leading term is a constant. The same scaling holds at $\alpha = 2$, since the Haldane-Shastry Hamiltonian share the same universality class of the Heisenberg antiferromagnet [30, 56]. By continuity, we argue the same

critical behavior extends to the whole intermediate region. This is in line with the Luttinger liquid conjecture (see Sec. 2.2). Since the latter is of interest for the subsequent analysis, we estimate its bounding *transition* points. From Fig. 3 is clear that there is a transition in parameter space at $\alpha = 0$.

We perform finite-size scaling on our data to estimate the critical value $\alpha_c$ of the JG wave functions separating a critical phase with respect to a Néel ordered state. This is a phenomenological finite-size scaling procedure, since it is inherently related to a parameter characterizing the *variational* wave functions, and not associated to a coupling term in a Hamiltonian. Nevertheless, it is useful to bound the region of validity of the reconstruction method (Sec. 3), which relies on relativistic invariance. We consider the scaled entanglement entropy $\tilde{S}(\alpha)$ as an order parameter, as well as its derivative:

$$\chi(\alpha) = \frac{d}{d\alpha}\tilde{S}(\alpha), \tag{16}$$

which is roughly a susceptibility. We choose to consider both these quantities since the scaling we have is very mild with system size. From Fig. 3, introducing $t = \log(L)$ and $\tilde{\alpha} = (\alpha - \alpha_c)/\alpha_c$, we use the following simplified scaling ansatz:

$$\chi(\alpha)t^\gamma = g(\tilde{\alpha}t^{1/\nu}), \tag{17}$$

$$\tilde{S}(\alpha)t^\beta = G(\tilde{\alpha}t^{1/\nu}). \tag{18}$$

To perform the finite size scaling we vary the exponents $\nu, \gamma$ and the critical value $\alpha_c$ over a suitable range of parameters. The fit is the best over different degrees of polynomials, test with a least-square method against the data [104]. By requiring the exponents to obey scaling relations $\gamma = \beta - 1/\nu$ we are able to reduce the fitting regime. We estimate the transition at $\alpha_c = 4.3 \pm 0.1$ with $\nu = 2.1 \pm 0.2$ and $\beta = -0.15 \pm 0.03$. Value and error bars are the average and standard deviations of the best fits varying the range of system sizes considered. In Fig. 4 we plot both the order parameters of interest and the optimal data collapse. While the quality of the collapses is generically good, the modest system size are not able to resolve more efficiently the exponent landscape, which results quite flat. We believe a more systematic analysis is needed to better characterize the entanglement entropy and its phase transition for the JG wave functions. This would be a useful test also for the Luttinger liquid conjecture in Ref. [65], where it is argued the transition is around the value $\alpha_c^{\text{conj}} = 4$. In this paper we choose to follow a more restrictive and cautious approach, focusing on subintervals of $\alpha \in (0,4)$ in the rest of the paper.

A concluding remark, which will be useful later, is about the $\alpha = 0$ point. As previously discussed in the context of participation spectrum, this point is peculiar since the JG state is in an equal-weight combinatorial superposition. Its exact entanglement entropy can be computed [80]:

$$S_{\alpha=0} = \frac{1}{2}\log\left(\frac{\pi L}{2}\right) + \frac{1}{2} - \log 2 + o(1). \tag{19}$$

We see that the pre-factor is different from the one in Eq. (15), signal that the state is not representative of the same phase. One can see this by investigating the properties of the exact parent Hamiltonian at $\alpha = 0$: the XXZ chain at the ferromagnetic transition [56,80]. This Hamiltonian has a gapless quadratic spectrum, thus it breaks relativistic invariance due to a different dynamical exponent[2] $z = 2$. This observation will be important when trying to reconstruct local Hamiltonians using a relativistic ansatz. Indeed, as we shall comment in Section 4, for $\alpha = 0$ the algorithm will not be able to return a correct parent Hamiltonian, as expected.

---

[2] This quantity measures the scaling ratio of space and time after a scale transformation. For relativistic theories $z = 1$.

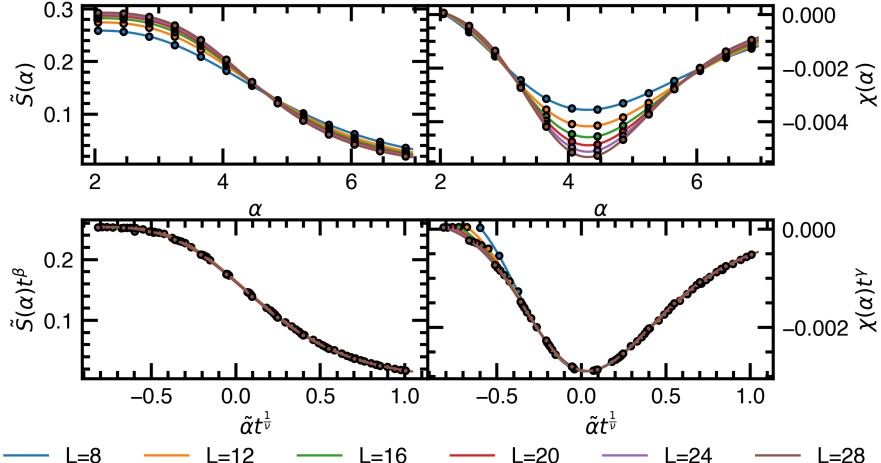

Figure 4: In the top panels we plot $\tilde{S}(\alpha)$ and $\chi(\alpha)$ nearby the expected transition $\alpha_c$. The results of the data collapse for the ansatz Eq. (17) are plotted in in the lower panels. The best fit gives an estimate of $\alpha_c = 4.3(1)$ while $\nu = 2.1(2)$, $\beta = -0.15(3)$. We consider the interval $\alpha \in (2,7)$ and $L = 8, 12, 16, 20, 24, 28$.

## 2.4 Correlation functions

To further characterize the Jastrow-Gutzwiller states, we compute the one-body and two-body spin correlation functions $\{\sigma^z, \sigma^+, \sigma^-\}$. Their scaling properties resolve the nature of the state being critical or not.

Due to the binary nature of the $n_i$ variables, for notational convenience we introduce the unary-not operator $F_{ij}$ acting on the site $i, j$, whose action on basis state is defined by logical negation on $n_i$ and $n_j$. Since the system exhibits a $U(1)$ symmetry related to number conservation, we compute only $U(1)$ invariant correlation functions. Recalling $\sigma^z = 2n - 1$ with $n$ the number operator we have:

$$
\begin{aligned}
\langle \sigma_i^z \rangle &= \sum_{\mathcal{P}_N(\{n\})} (2n_i - 1) \big| \psi_\alpha(\{n\}) \big|^2, \\
\langle \sigma_i^z \sigma_j^z \rangle &= \sum_{\mathcal{P}_N(\{n\})} (2\delta(n_i, n_j) - 1) \big| \psi_\alpha(\{n\}) \big|^2, \\
\langle \{\sigma_i^+, \sigma_j^-\} \rangle &= \sum_{\mathcal{P}_N(\{n\})} (1 - \delta(n_i, n_j)) \psi_\alpha(\{n\}) \psi_\alpha(F_{ij}\{n\}).
\end{aligned}
\tag{20}
$$

At half-filling the first one is identically zero. The latter ones can be easily implemented numerically. The correlation length can be extrapolated through finite size scaling of the connected correlation function $\langle \sigma_i^z \sigma_{i+L/2}^z \rangle_c$:

$$
\langle \sigma_i^z \sigma_j^z \rangle_c \equiv \langle \sigma_i^z \sigma_j^z \rangle - \langle \sigma_i^z \rangle \langle \sigma_j^z \rangle = a \frac{e^{|i-j|/\xi}}{|i-j|^\gamma},
\tag{21}
$$

$$
\frac{1}{\xi} = -\lim_{L \to \infty} \frac{\log\left( \langle \sigma_i^z \sigma_{i+L/2}^z \rangle_c \right)}{L/2} \equiv \lim_{L \to \infty} \frac{1}{\xi_L}.
\tag{22}
$$

Here $a$ is a constant, while $\gamma$ characterize the algebraic decay. In all the above equations, we exploited periodic boundary conditions.

Let us stress that the definition Eq. (22) is meaningful only when the cluster decomposition principle holds. This requires the connected correlation function to decay to zero with

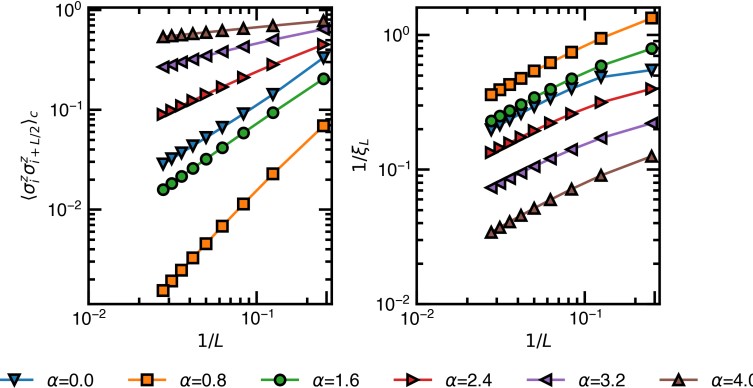

Figure 5: (Left) Connected correlation function against $1/L$. The seemingly algebraic decay suggests the the cluster decomposition requirement is fulfilled for the considered $\alpha$. These are chosen representatives of the critical phase. (Right) Inverse correlation length for different values of $\alpha$ in the critical phase. For both plots, we considered chains of lengths $4 \leq L \leq 36$. To avoid odd/even effects, we present only $L$ multiples of four.

the distance between the spins. This definition is used throughout in the literature of critical phenomena, where the phase is defined through the ground state manifold of specific Hamiltonians [82]. In this context, symmetry broken phases at finite system size manifest themselves as a coherent superposition of the ground states in the different symmetry sectors (GHZ states) [23]. The latter are a remarkable example of states which do not respect the cluster decomposition.

Having the above remark in mind, we consider the definition Eq. (22) to also characterize the parameter space of the JG wave functions. Here we first check the system is fulfilling the cluster decomposition principle condition. When this is not the case, we expect the JG state to be representative of a finite size symmetry broken phase. Within this setting, if the parameter $\xi$ is finite, the exponential behavior dominates on the algebraic one and the system is gapped, while if $\xi \to \infty$ the system behaves as critical.

In Fig. 5, we show the results of our fitting procedure, plotting the inverse correlation length versus $1/L$. For the chain lengths considered, the thermodynamic limit is difficult to estimate since at finite size the inverse correlation length $1/\xi_L$ can be trusted upon the value $1/L$. However, all values $\alpha < 4.0$ are compatible with an infinite correlation length.

For large positive values and negative values of $\alpha$, the cluster decomposition principle fails. The corresponding GHZ states (introduced in Sec. 2.2), representatives of symmetry broken phases, are confirmed to reproduce the correlation functions of the JG wave functions. A detailed discussion is given in Appendix A.

# 3 Entanglement guided search for parent Hamiltonians

In this section we summarize the scheme we employ to reconstruct parent Hamiltonians [50]. As previously remarked, this method requires additional conditions to work. This in contrast to other techniques [45, 46] based on the quantum covariance matrix (QCM). The latter are simpler to implement since are based on requiring the input state to satisfy the zero energy variance condition. Thus, those methods generically guarantee that the input state is an eigenstate (not the ground state) of the parent Hamiltonian. Here comes the reason we have chosen to use the Bisognano-Wichmann Ansatz (BWA) scheme: the additional physical constraints guar-

antee the parent Hamiltonian of the input state as the *ground state*. This condition is at the core of eventual simulation protocols, since excited state are less robust in analogue experiments. Nevertheless, the relativistic requirement can be applied only to a narrow number of settings: for example if non-translational system are considered, such as disordered systems, BWA fails while QCM still gives meaningful results [103], provided a *a fortiori* analysis is done on the parent Hamiltonian space and their spectra. The method we adopt is based on the Bisognano Wichmann (BW) theorem, which for convenience we recap in the first subsection. Then, we introduce the common ingredients shared with other aforementioned techniques [45–47, 49, 50]. We conclude this section by presenting the algorithm and our chosen implementation.

## 3.1  Bisognano-Wichmann theorem and lattice models

By definition, reduced density matrices are positive operators with bounded spectrum $\sigma(\rho_A) \subset [0, 1]$. Consequently, it is always possible to find a lower bounded operator $K_A$ such that $\rho_A \sim \exp(-K_A)$. This object is usually referred to as entanglement or modular Hamiltonian, and in general is highly non-local, being the logarithm of the non-local operator $\rho_A$.

Remarkably, Bisognano and Wichmann proved that the entanglement Hamiltonian acquire a local density when considering the ground state of a relativistic quantum field theory partitioned into two half-spaces [52, 53, 77, 81]. Moreover, the density of this modular operator is proportional to the one of the theory Hamiltonian. The statement is the following.

**Theorem (Bisognano Wichmann)**  Given a local relativistic QFT in $d + 1$ spacetime dimensions, described by an Hamiltonian $H = \int d^d x \mathcal{H}(x)$ the half-space reduced density matrix of the vacuum $|\Omega\rangle$ is:

$$\rho_A = \mathrm{tr}_B |\Omega\rangle \langle\Omega| = \frac{e^{-(2\pi/v)K_A}}{Z_A}, \tag{23}$$

$$K_A = \int_A d^d x \, x_1 \mathcal{H}(x), \qquad Z_A = \mathrm{tr}_A \rho_A. \tag{24}$$

Here $A$ and $B$ are respectively the manifolds $A = \{x \in \mathbb{R}^d : x_1 \geq 0\}$ and its complementary, while $v$ is the sound velocity of the relativistic excitations. Sometimes, the pre-factor $\beta \equiv 2\pi/v$ is dubbed entanglement temperature due to the analogy with respect to thermal density matrices.

More recently, this result has been revisited in the context of holography and many-body physics [83–94]. In particular, the theorem has been extended for theories with conformal invariance [83, 85, 86]. Given the subsystem $A = \{x \in \mathbb{R}^d | 0 \leq r \leq R, r = ||x||\}$, its entanglement Hamiltonian reads:

$$K_A = \int_A d^d x \, r \left(1 - \frac{r}{R}\right) \mathcal{H}(x). \tag{25}$$

Interestingly, when considering lattice systems exhibiting relativistic low-lying excitations, the discretisation of Eq. (23) and Eq. (25) gives a fine approximation of their reduced density matrices [78, 93, 95–100], with even exact results for specific models [101, 102]. Moreover, the discrepancies due to the lattice structure disappear in the thermodynamic limit.

This motivates the core idea behind the BWA method: to find optimal BW entanglement Hamiltonian describing the reduced density matrix of state of interest, in our case the Jastrow-Gutzwiller wave functions. For concreteness, in the remaining of this paper we make use of

the discrete version of Eq. (25) in 1D system of size $L$ and $A = \{1, 2, \ldots, L/2\}$:

$$\rho_A^{\text{BW}} = \frac{e^{-K_A}}{Z_A}, \qquad Z_A = \text{tr}_A \rho_A, \qquad H = \sum_{r=1,2,\ldots,L/2} h_r, \tag{26}$$

$$K_A = \sum_{r=1,2,\ldots,L/2} r \left(1 - \frac{2r}{L}\right) h_r. \tag{27}$$

Here $r$ label the sites, $h_r$ is the lattice density of the Hamiltonian $H$, while $K_A$ the corresponding modular operator. Conventionally, we chose to absorb the entanglement temperature in the Hamiltonian density couplings $h_r$.

## 3.2 Basis of local operators

To quantitatively describe the theory and entanglement Hamiltonians on the lattice we introduce the basis of local operators. As previously mentioned, these fully characterize the operator space of the parent Hamiltonian search.

We say an operator is $k$-local if either (1) it has finite domain $k$-nearby few body operators, or (2) it is written as a linear combination of the latter. Furthermore, we require $k$ to be constant for any finite system size $L$ we consider. If these conditions are not fulfilled, we say the operator is non-local.

We define a basis of $k$-local operators as the set of matrices $\{O_{\mu,r}\}_{\mu \in I, r \in \Gamma}$. Here $I$ is a set of internal indices, while $\Gamma \subset \Lambda$ is a set of sub-lattice ones. Depending on the values of $I$ and $\Gamma$, these basis span different vector spaces of local operators, whose generic element is:

$$H = \sum_{\alpha \in I, r \in \Gamma} w_{\alpha,r} O_{\alpha,r}. \tag{28}$$

The dimension of these spaces is thus given by the combined cardinality of the label sets $D = |I||\Gamma|$.

Before moving on, we clarify the above notation through few examples. Let us first consider the Pauli algebra at each site $r \in \Gamma = \Lambda$:

$$\mathcal{B}_1 = \{1_r, \sigma_r^x, \sigma_r^y, \sigma_r^z\}_{r \in \Lambda} \quad \text{with} \quad O_{0,r} = 1_r, \ O_{1,r} = \sigma_r^x, \ O_{2,r} = \sigma_r^y, \ O_{3,r} = \sigma_r^z. \tag{29}$$

The generic linear combination is:

$$H = \sum_{r \in \Lambda} \sum_{\alpha=0}^{3} w_{\alpha,r} O_{\alpha,r}. \tag{30}$$

We see the total dimension is $D = 4L$ in this case. A less trivial example is the two-body nearest neighboring interactions:

$$\mathcal{B}_2 = \mathcal{B}_1 \cup \{\sigma_r^x \sigma_{r+1}^x, \sigma_r^y \sigma_{r+1}^y, \ldots, \sigma_r^z \sigma_{r+1}^z\}_{r \in \Lambda}. \tag{31}$$

Here $\alpha$ covers, in addition to the elements in Eq. (31), the following two-body operators at each site $r$:

$$O_{4,r} = \sigma_r^x \sigma_{r+1}^x, \ O_{5,r} = \sigma_r^x \sigma_{r+1}^y, \ldots, \ O_{10,r} = \sigma_r^z \sigma_{r+1}^y, \ O_{11,r} = \sigma_r^z \sigma_{r+1}^z. \tag{32}$$

The linear space has dimension $D = 12L$. Imposing symmetries one can reduce the dimension $D$ of the operator space, in the same fashion symmetry constraints can be used to block

diagonalize observables. For example, imposing $U(1)$ and translational symmetry, a possible operator basis is the following:

$$\mathcal{B}_{NN(2)} = \left\{ \sum_{r\in\Lambda}(\sigma_r^+\sigma_{r+1}^- + \sigma_r^-\sigma_{r+1}^+), \sum_{r\in\Lambda}(\sigma_r^z\sigma_{r+1}^z), \sum_{r\in\Lambda}\sigma_r^z \right\} \equiv \{h_1, h_2, h_3\}. \tag{33}$$

Here, the index $\alpha$ takes three values ($D = 3$) and the Hamiltonian is:

$$H = \sum_\alpha w_\alpha h_\alpha \equiv \sum_\alpha w_\alpha \left( \sum_{r\in\Lambda} O_{\alpha,r} \right). \tag{34}$$

In the second step of the above equation, we wrote the operators $h_\alpha$ in terms of Eq.(31). Thus, the freedom of choosing the operator basis enables us to specify the required symmetries of the parent Hamiltonian, and it allows a reduction of complexity (for translational invariant systems, $D \sim \mathcal{O}(1)$ in system size).

Motivated by the symmetries of the JG states, we will consider the following basis for $k \geq 2$:

$$\mathcal{B}_{NN(2)} = \left\{ \sum_{r\in\Lambda}(\sigma_r^+\sigma_{r+1}^- + \sigma_r^-\sigma_{r+1}^+), \sum_{r\in\Lambda}(\sigma_r^z\sigma_{r+1}^z), \sum_{r\in\Lambda}\sigma_r^z \right\}, \tag{35}$$

$$\mathcal{B}_{NN(k+1)} = \mathcal{B}_{NN(k)} \cup \left\{ \sum_{r\in\Lambda}(\sigma_r^+\sigma_{r+k}^- + \sigma_r^-\sigma_{r+k}^+), \sum_{r\in\Lambda}(\sigma_r^z\sigma_{r+k}^z) \right\}.$$

Varying the value of $k$ we consider an increasing number of nearest-neighboring hopping and exchange operators. Finally, since the physics of the JG state at $\alpha = 2$ is captured by a long range model, we shall consider the basis of non-local operators:

$$\mathcal{B}_{\mathrm{LR}} = \left\{ \sum_{r<m\in\Lambda} \frac{\pi^2}{L^2}\frac{\sigma_r^+\sigma_m^- + \sigma_r^-\sigma_m^+}{\sin^2(\pi(r-m)/L)}, \sum_{r<m\in\Lambda} \frac{\pi^2}{L^2}\frac{\sigma_r^z\sigma_m^z}{\sin^2(\pi(r-m)/L)} \right\}. \tag{36}$$

These basis are both $U(1)$ and translationally invariant, thus exhibits coefficients $w_\alpha$ not depending on lattice sites. In literature, non-translational invariant basis have been employed in the reconstruction of disorder system Hamiltonians [45, 48, 103], or to enlarge the set of Hamiltonians having the input state as an eigenstate [46].

## 3.3 Parent Hamiltonian reconstruction method

We are now in position to present the BWA scheme. Let $\rho_A^{\mathrm{input}}$ be the half-system reduced density matrix of the the input state. We want to find optimal coefficients $w_\alpha$ in Eq. (34) such that:

$$\rho_A^{\mathrm{input}} \simeq \rho_A^{BW}(\{w_\alpha\}). \tag{37}$$

This optimization can be implemented using any estimator of distance between $\rho_A^{\mathrm{input}}$ and the model reduced density matrix $\rho_A^{BW}(\{w_\alpha\})$. For example one can use the Kullback-Leibler divergence between the participation spectra of the reduced density matrices [60]. This estimator has the advantage of being easy to implement even for larger spacetime dimensions, but has the drawback of leading in general to a non-convex optimization. Such obstacle can be anyway surpassed using stochastic optimization algorithms. Instead, for the class of models described by the basis in Eq. (35) and Eq. (36), it can be proven that any convex estimator acting on the space of density matrices leads to a convex optimization problem (with a unique solution). Among these, we have found particularly useful for numerical implementations the

relative entropy, which we adopt in the remaining of this paper. Given two density operators $\rho$ and $\sigma$, it is defined as:

$$S(\rho|\sigma) = \mathrm{Tr}(\rho \log \rho) - \mathrm{Tr}(\rho \log \sigma). \tag{38}$$

This function quantifies the distance between between $\rho$ and $\sigma$, it is non-negative $S(\rho|\sigma) \geq 0$ (with the equality holding only if $\rho = \sigma$) and it is jointly convex. In particular, its restriction to a single argument is a convex function. As already stated, the relative entropy leads to a convex optimization admitting, up to numerical precision, a *unique* solution [50]:

$$\vec{w}^\star = \arg\min_{\vec{w}} S(\rho|\sigma_{\mathrm{BW}}(\vec{w})). \tag{39}$$

The relative entropy value express a "distance" in the reduced density matrix manifold, and quantify the difference between the initial wave function and the closer one fulfilling the BW theorem.

We implement a gradient descent on the relative entropy. Introducing the notation $\partial_\alpha = \partial/\partial w_\alpha$ and:

$$\langle O \rangle_{\mathrm{GS}} \equiv \mathrm{Tr}(O\rho_A), \quad \langle O \rangle_{\mathrm{BW},\vec{w}} \equiv \mathrm{Tr}(O\rho_A^{\mathrm{BW}}(\vec{w})), \tag{40}$$

the gradient of the relative entropy reads

$$\partial_\alpha S(\rho_A|\rho_A^{\mathrm{BW}}(\vec{w})) = \langle h_\alpha \rangle_{\mathrm{GS}} - \langle h_\alpha \rangle_{\mathrm{BW},\vec{w}^{(n)}}. \tag{41}$$

We remark that the actual input needed are just the expectation values over the ground state and over the "thermal" BW density matrix. The former can be sometimes computed analytically, as in the JG states (see Section. 2), while the latter can be implemented with different numerical methods, including quantum Monte Carlo when no sign problem is present.

# 4 Reconstruction of Jastrow-Gutzwiller parent Hamiltonians

In this section, we apply the entanglement based reconstruction technique to JG wave functions, considering different choices for the operator basis. We quantify the quality of the reconstruction utilizing (1) relative entropies between reduced density matrices, (2) wave function overlaps, and (3) correlation functions. In view of the discussion in section Sec. 2, we focus here on the regime $0 < \alpha < 4$; the regimes where the wave functions are captured by GHZ states are instead discussed in Appendix A.

## 4.1 Models for reconstruction

We consider two paradigmatic classes of operators as candidates for the parent Hamiltonian reconstruction. The first one are the $k$-local Hamiltonians constructed from the basis $\mathcal{B}_{NN(k)}$ introduced in Eq. (35):

$$H_k = \sum_r \sum_{p=1}^{k-1} \frac{J_p}{2}(\sigma_r^+ \sigma_{r+p}^- + \mathrm{h.c.}) + \Delta_p \sigma_r^z \sigma_{r+p}^z + h\sigma_r^z. \tag{42}$$

These Hamiltonians for $k \leq 4$ are archetypal for the study of strongly correlated matter in 1D and 2D, and have been used for *ab initio* numerical studies of quantum spin liquid phases in different lattices [18–21, 51, 104]. We notice that these operators contains the XXZ and the $J_1 - J_2$ model as particular cases. The second class are long-range XXZ Hamiltonians constructed from the basis $\mathcal{B}_{\mathrm{LR}}$ in Eq. (36):

$$H_{\mathrm{LR}} = \frac{\pi^2}{L^2} \sum_{r<m} \frac{1}{\sin^2(\pi(m-r)/L)} \left( \frac{J_1}{2}(\sigma_r^+ \sigma_m^- + \mathrm{h.c.}) + \Delta_1 \sigma_r^z \sigma_m^z \right). \tag{43}$$

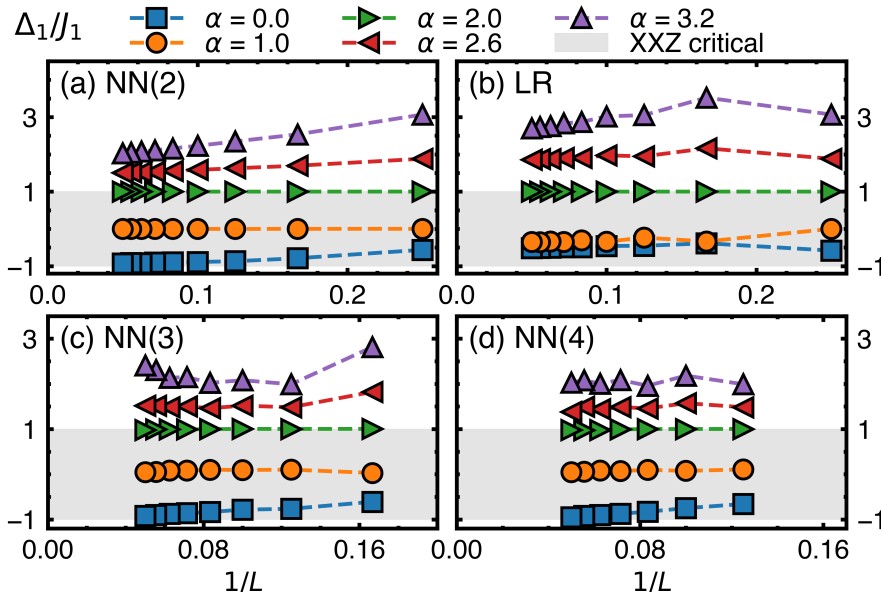

Figure 6: Scaling of the ratio of the converged coefficient $\Delta_1/J_1$ over different basis for different values of the parameter $\alpha$. The shaded region corresponds to the critical values of the XXZ chain.

The reason in the latter choice is twofold: on one hand $J_1 = \Delta_1$ is the Haldane-Shastry Hamiltonian, the exact parent Hamiltonian at $\alpha = 2$. On the other hand, in Ref. [54] Shastry conjectured that $\alpha \neq 2$ is the ground state of Eq. (43). We remark that the parent Hamiltonian is defined up to an overall multiplicative constant which sets the energy scales, and an additive zero energy value. Thus, without loss of generality, we factor out the $J_1$ term and we are interested in the values $\{w/J_1\}$.

**Numerical implementation** We search parent Hamiltonians of the above form through the BWA technique. The implementation is based on exact diagonalization (ED) routines in Fortran, using standard libraries and LAPACK [105]. We performed gradient descents with various threshold error $\epsilon_{\text{th}} = 10^{-3} - 10^{-6}$. In the considered region, we notice no qualitative change in the observable behavior, although a smaller threshold error requires more steps in the gradient descent convergence. For convenience, we present the results only for $\epsilon_{\text{th}} = 10^{-4}$. At this value, the observables are determined with a precision of around 0.1%.

The initial value of the couplings is drawn by a uniform random distribution on the interval $[-2, 2]$. Here the spreading plays a minimal role: since the optimal solution is unique (see Section 3), the only ambiguity is numerical and due to the truncation to $\epsilon_{\text{th}}$. The resulting uncertainty is in the last sensible digit of the relative entropy and of the other observables, which we lift through averaging over 50 initial configurations. As argued in Sec. 2, in the thermodynamic limit the system should exhibit a critical regime in the region $\alpha \in (0, 4.30)$. However, for the modest values considered $L \in \{4, 6, \ldots, 20\}$, we chose to focus on the subregion $\alpha \in (0, 4)$, where finite-size effects are less severe.

## 4.2  Diagnostics for reconstruction

Let us introduce the observables we use to access the quality of the parent Hamiltonian reconstruction. Firstly, we evaluate the relative entropy $S(\rho_{\text{jas}}|\rho_{BW})$ between the converged BW reduced density matrix $\rho_{BW}$ and the exact JG one $\rho_{\text{jas}}$. Since this function is a "distance" in the

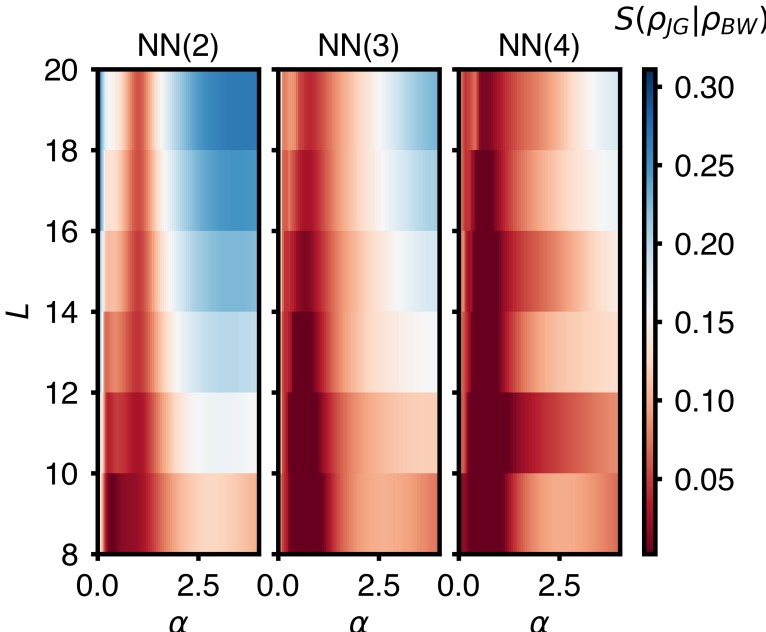

Figure 7: Relative entropy of the JG reduced density matrix and the BW converged one. We see that enlarging the domain of the operator involved, the quality of the results increases. The line $\alpha = 1$ corresponds to a free fermions gas.

density matrix space, it quantifies how much the BW density matrix approximates the input state.

We then introduce the module of the overlap $|\langle \psi_{\text{jas}} | \psi_{\text{rec}} \rangle|$ between the JG wave function $|\psi_{\text{jas}}\rangle$ and the ground state of the reconstructed Hamiltonian:

$$H_{\text{rec}} | \psi_{\text{rec}} \rangle = E_{GS} | \psi_{\text{rec}} \rangle. \tag{44}$$

We stress that this quantity is meaningful only for finite size systems, since it decays to zero in the thermodynamic limit, for any arbitrary small difference between two vector states (in analogy with orthogonality catastrophe [106]).

Finally, we compute the following quantity, a *cumulative* estimate of how much the correlation functions over the reconstructed state differ from the exact ones:

$$V(\text{rec}|\text{jas}) \equiv \frac{1}{\sqrt{L}} \left\| \langle \sigma_0^z \sigma_j^z \rangle_{\text{rec}} - \langle \sigma_0^z \sigma_j^z \rangle_{\text{ex}} \right\| = \frac{1}{\sqrt{L}} \sqrt{\sum_{j=1}^{L-1} \left( \langle \sigma_0^z \sigma_j^z \rangle_{\text{rec}} - \langle \sigma_0^z \sigma_j^z \rangle_{\text{ex}} \right)^2}. \tag{45}$$

Here the first term is the correlation function respectively on the ground state of the reconstructed parent Hamiltonian and on the JG state eq (20). The $1/\sqrt{(L)}$ factor renders this object non-extensive, which is desirable when comparing different system sizes. For convenience, we call this operator the cumulative correlation difference.

Equipped with these tools, in the following subsections we separately present the analysis for the previously introduced basis Eq. (42) and Eq. (43). On the former, we first discuss overlaps and relative entropies for different basis choices, and finally discuss correlation functions. On the latter, we focus the analysis only on the relative entropy.

## 4.3   Reconstruction with $NN(k)$

We begin by considering the models in Eq. (42) for $k = 2, 3, 4$. If a $p$-local Hamiltonian exists, we expect the terms $k > p$ to be finite size terms and to decay to zero enlarging the system

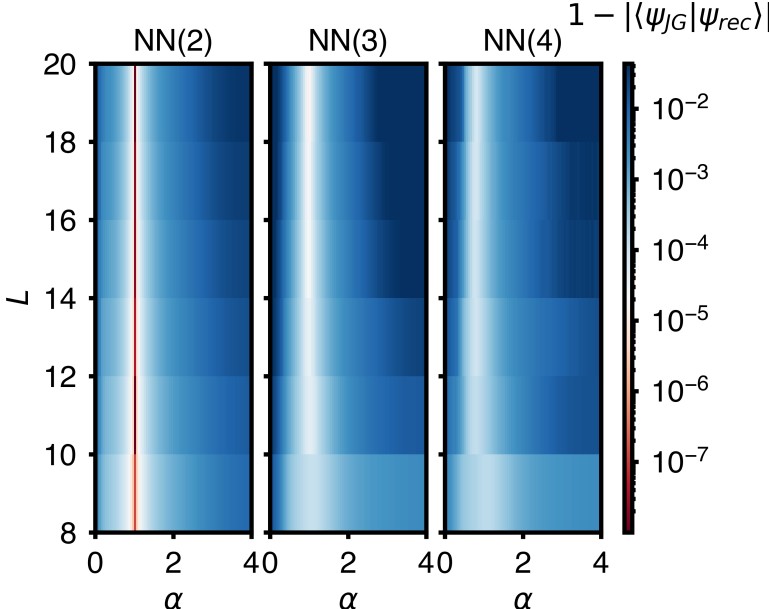

Figure 8: Overlap matrix between the ground state of the reconstructed parent Hamiltonian and the input JG state. As in Fig. 7, NN(k) labels the model used. (Left) Using only nearest neighbors interactions, the reconstruction is faithful at $\alpha = 1$. However, enlarging the nearest neighboring operators and fixing the error at $\epsilon \sim 10^{-4}$ the superposition is susceptible to finite size effects, as shown in panel (Center),(Right).

size. We anticipate that our result suggests that an exact local parent Hamiltonian exists only for $\alpha = 1$ (see, e.g., the scaling of the overlap depicted in Fig. 8), which corresponds to free fermions 2-local Hamiltonian. At different values of $\alpha$, the reconstruction is only approximate, although it improves considerably increasing the basis $NN(k)$. We deduce that the exact parent Hamiltonian should involve long-range interactions.

**Search for nearest-neighbor Hamiltonians. -** Let us first restrict the easiest setting, that is choosing the $NN(2)$ basis. In this case, the Hamiltonian Eq. (42) corresponds to the XXZ model. The value of interest is $\Delta_1/J_1$. When this is zero, the model reduces to the XX chain, which is a free fermion model up to a Jordan Wigner transformation. Moreover, it is interesting to compare our results with those of Ref. [56]. There, the authors considered the inverse variational problem, optimizing the parameter $\alpha$ with respect to the fixed ratio of $\Delta_1/J_1$. They argue that for $\alpha \in [0, 2]$ the wave functions are representatives of the critical phase $\Delta_1/J_1 \in [-1, 1]$ characterizing the spin-1/2 XXZ chain. Our results are compatible with their findings and the analytic results (Fig. 6).

For larger values of $\alpha$, our results still indicate a very clear convergence to the thermodynamic limit. Moreover, the extrapolated values (Table 1) always indicate that $\Delta_1 > J_1$ in this regime: this is compatible with an antiferromagnetic state with a very large correlation length. This finding is highly non-trivial, as there is no guarantee that our method shall return the correct parent Hamiltonian even in the presence of strong finite-volume effects, that have to be expected in this regime since, in the XXZ model, the transition to an antiferromagnetic phase belongs to the Berezinskii-Kosterlitz-Thouless universality class.

Table 1: Converged couplings using the $NN(2)$ basis for different $\alpha$ and $L$. Here we present the results for an unconstrained optimization in order to benchmark the algorithm. The data indicate the symmetry is always preserved, while the relative ration of the couplings are consistent with the exact cases: $\alpha = 1$ with the $XX$-chain and $\alpha = 2$ with the Haldane-Shastry Hamiltonian.

| $\alpha$ | L | $h$ | $\Delta_1$ | $J_1$ |
|---|---|---|---|---|
| 0.2 | 12 | 0.0000 | -4.4250 | 5.4065 |
| 0.2 | 16 | 0.0000 | -4.3136 | 5.1752 |
| 0.2 | 20 | 0.0000 | -3.9237 | 4.7028 |
| 1.0 | 12 | 0.0000 | -0.0014 | 1.7919 |
| 1.0 | 16 | 0.0000 | -0.0006 | 1.7751 |
| 1.0 | 20 | 0.0000 | -0.0004 | 1.7636 |
| 2.0 | 12 | 0.0000 | 1.1066 | 1.1066 |
| 2.0 | 16 | 0.0000 | 1.0823 | 1.0823 |
| 2.0 | 20 | 0.0000 | 1.0477 | 1.0576 |
| 2.8 | 12 | 0.0000 | 1.5227 | 0.8700 |
| 2.8 | 16 | 0.0000 | 1.4258 | 0.8371 |
| 2.8 | 20 | 0.0000 | 1.3371 | 0.7973 |

**Search beyond nearest-neighbor Hamiltonians. -** It is important to test the stability of these findings both with respect to enlarging the basis, considering $NN(k > 2)$, and to system size. We thus considered the reconstruction also $NN(3)$ and $NN(4)$, and studied the behavior of the couplings $\{w_\alpha/J_1\}$. As shown in Fig. 7 and Fig. 8, both the relative entropy and the overlap improve including higher-$k$ terms. In addition, the magnitude of the couplings corresponding to the latter seems to increase with system size (see Fig. 9), suggesting that the exact Hamiltonians for the Jastrow-Gutzwiller states are long-ranged. An exception is the point $\alpha = 1$, whose reconstructed Hamiltonian converges to the XX chain. As argued in Sec. 2, this is expected due to analytic arguments.

**Ferromagnetic JG wave function. -** Another particular point is $\alpha = 0$. There, the corresponding JG wave function is the exact ground state of the ferromagnetic transition point XXZ. The BWA in principle should not work being this point described by a non-relativistic field theory [80]. However, the converged coupling is flowing toward the correct $\Delta_1/J_1 = -1$ enlarging the system size. Importantly, this result is strongly dependent on the basis chosen, and we see that it is unstable adding larger hopping terms ($NN(3)$ and $NN(4)$). Here the modulus of the couplings corresponding to ($k > 2$)-local terms increases, signal that a *relativistic* exact parent Hamiltonian for this point, if it exists, it is strongly long-range.

**Correlation functions. -** Finally, we present in Fig. 11 the results for the cumulative correlation difference $V(\text{rec}|\text{jas})$. At fixed system size $L$, it slightly increases when including higher $k$-terms. This is counterintuitive, since we observe that a larger basis $NN(k)$ leads to states that are more similar to the JG wave functions (see Fig. 7 and Fig. 8). With the present analysis, we are not able to fully characterize if this trend is due to finite size effects or it has a more systematic nature. A possible explanation would be hidden in the BWA algorithm: since it optimizes over the short-$k$ correlations (see Eq. (41)), the large distance correlators are less controlled and are subject to frustration effects. Within this interpretation, these discrepancies may suggest that longer range terms are required in the optimization to faithfully reconstruct an exact parent Hamiltonian.

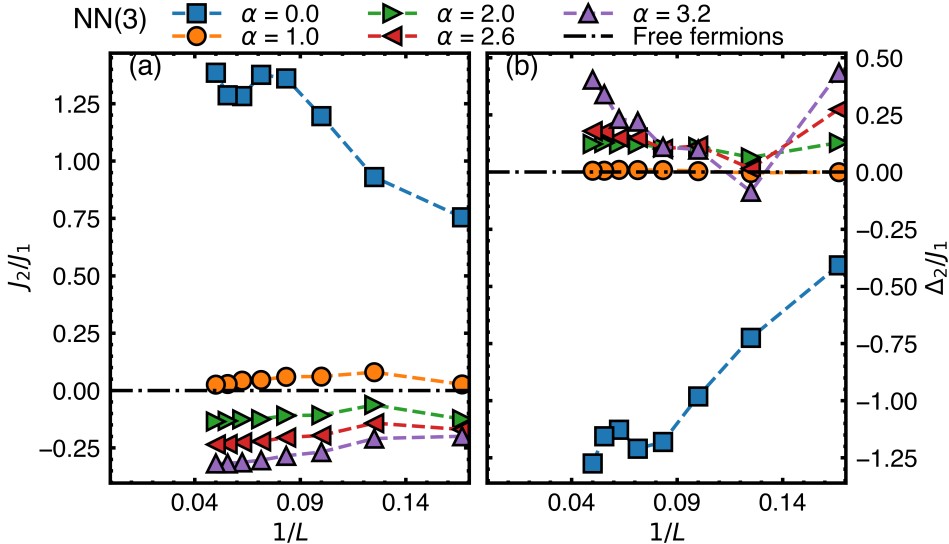

Figure 9: Ratios of the converged couplings $\Delta_2/J_1$ and $J_2/J_1$ versus inverse system size. We see that $\alpha = 1$ is flowing toward the XX Hamiltonian (see also Fig. 6), while the other converged values are stationary in non-null values, suggesting long range 2-body physics for the JG states.

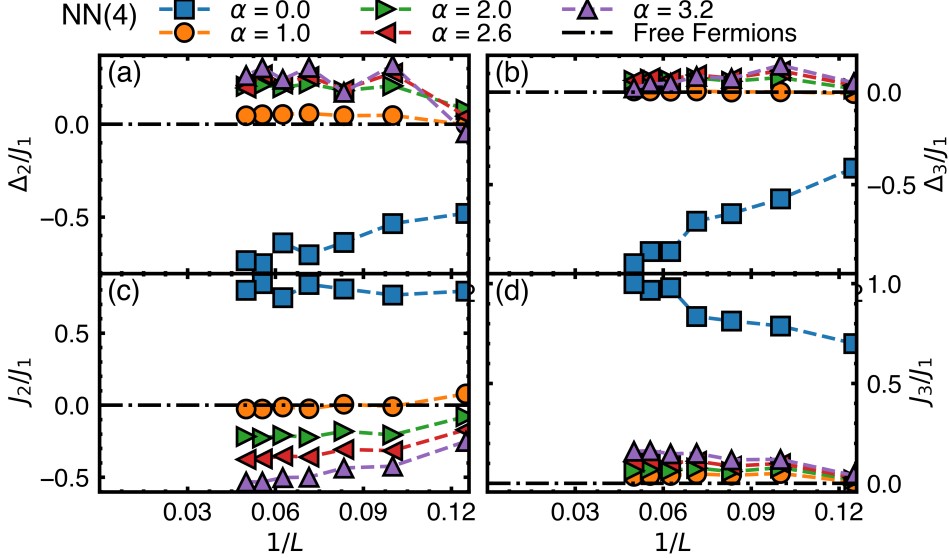

Figure 10: Ratios of the converged couplings ratios versus inverse system size using the $NN(4)$ basis. As for the previous cases, we see that $\alpha = 1$ is flowing toward the XX Hamiltonian (see also Fig. 6, Fig. 9). Other values of $\alpha$ suggest a long range 2-body physics for the JG states.

Instead, at a fixed value of $k$, the cumulative correlation difference seems to saturate at some finite value. Being such an object deviation measure from a standard value (see Eq. (45)), it roughly gives how much on percentage the correlation functions change at a fixed site. In the worse scenario of our results, this has a value of around 10%. One may compare our findings with the exact results of the Haldane-Shastry model and the antiferromagnetic Heisenberg

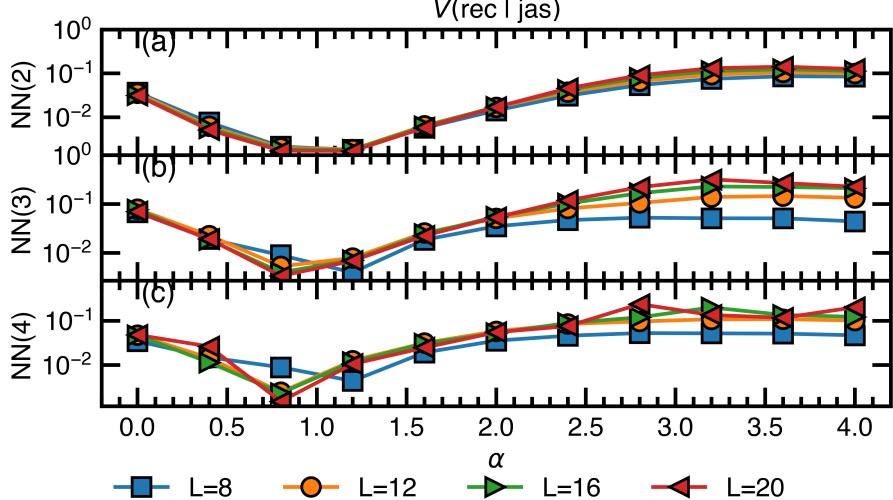

Figure 11: Norm of the cumulative correlation difference $V(\text{rec}|\text{jas})$, defined in Eq. (45), for different chain lengths $L$.

chain [30, 31, 107]:

$$\langle \sigma_i^z \sigma_j^z \rangle = \begin{cases} j & \text{JG}(\alpha = 2) & \text{XXZ} \\ 1 & -0.5894 & -0.5908 \\ 2 & 0.225706 & 0.2427 \end{cases} . \tag{46}$$

From the latter equations, we read the relative error of the nearest neighboring correlators and next-nearest neighboring ones, respectively of 2% and of 8%.

Combining the above reasonings, we state the reconstructed parent Hamiltonians are only approximate and the true parent Hamiltonians for the JG states require non-local terms. This further confirms our previous analysis. The exception is the point $\alpha = 1$, where the cumulative correlation difference improves both with system size and by including larger $NN(k)$.

**Relative error of the variational energy -**    As a last check we compute the variational energy of the parent Hamiltonian with respect to the Jastrow-Gutzwiller input state:

$$E_{\text{var}} = \langle \psi_{\text{jas}} | H | \psi_{\text{jas}} \rangle, \tag{47}$$

and compare with the exact ground state energy $E_{\text{gs}}$. The results are quantitatively compared via the relative error:

$$\text{err} = \frac{|E_{\text{var}} - E_{\text{gs}}|}{|E_{\text{gs}}|}. \tag{48}$$

We present the our results in figure Fig. 12. At fixed value of $NN(k)$, our data suggest a mild linear growth of the relative error with system size. A linear extrapolation of the thermodynamic limit is given. All the considered cases lie within 1% of relative error in the energy landscape. Interestingly, at fixed $L$ the relative error increases including larger $NN(k)$, in a similar fashion to what we observe in the correlation functions. At present we cannot fully understand and characterize such counterintuitive behavior. As already mentioned in the previous paragraph, this may be due to the algorithm forcing the optimization on a finite size landscape and creating frustration effects. The latter likely explain the case $\alpha = 2$, which should converge to the Haldane-Shastry pre-factors. Another possibility is that a new operator content is needed, and the chosen basis cannot grasp the thermodynamic properties of the systems. Further investigations on this problem are left for future studies.

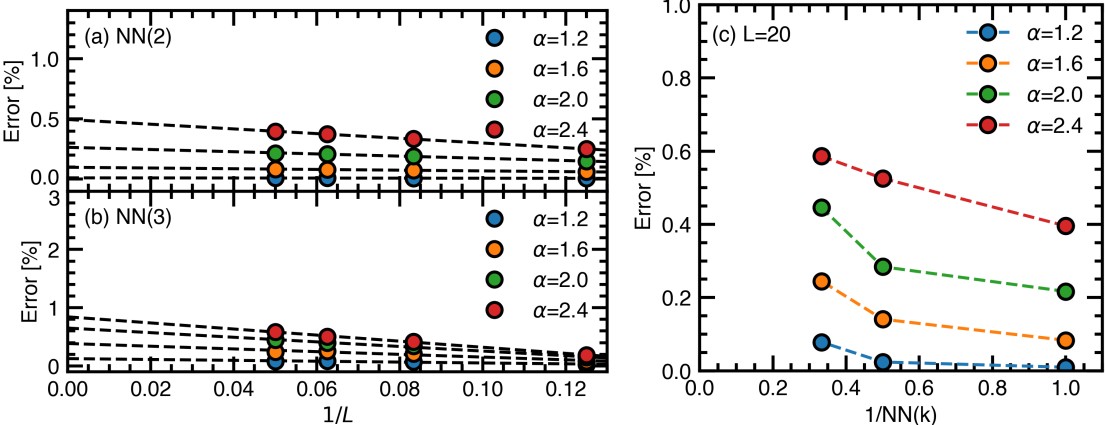

Figure 12: Relative error (in percentage) as a function of $1/L$ on the parent Hamiltonian constructed from the $NN(2)$ basis (panel (a)) and from the $NN(3)$ basis (panel (b)). The figure shows a very modest error even in the thermodynamic limit. (c) Relative error (in percentage) as a function of $1/k$, the number of nearest neighbors included in the reconstruction basis. At present we cannot infere a clear thermodynamic behavior for $k \to \infty$, as the number of points is too modest to fit. Nonetheless, the error seems bounded within 1%-2%.

### 4.4 Reconstruction with the long range model

We investigate the reconstruction when considering the model Hamiltonian Eq. (43), limiting our discussion to the relative entropy detector (see Sec. 4.2). The couplings are reported in Fig. 6, compared with the $NN(k)$ cases. For the chain lengths considered, only at $\alpha = 2$ the relative entropy shows a decreasing trend with system size (Fig. 13). This indeed corresponds to the exact Haldane-Shastry parent Hamiltonian. However, except at this fine-tuned point, the relative entropy grows with system size, suggesting the parent Hamiltonian Eq. (43) is no the exact parent Hamiltonian for $\alpha \neq 0$, and other more intricated terms must be added.

## 5 Conclusion and outlooks

In this work, we reconstructed approximate parent Hamiltonian for the one-dimensional Jastrow-Gutzwiller wave functions. We identified a region in parameter space where these wave functions display critical properties. Outside this interval, they are effectively described by Schrödinger cat states. Most likely, they are representatives of symmetry broken phases and their parent Hamiltonian is classical and constrained by the half-filling condition on the states.

For the reconstruction technique, first we considered $k$-local Hamiltonians. We confirm the exact point $\alpha = 1$ corresponding to free fermions, obtaining the XX Hamiltonian. At $\alpha = 0$ the method fails to find local and relativistic parent Hamiltonians. This is due to a breakdown in the relativistic invariance in the wave function, whose exact parent Hamiltonian manifest gapless quadratic spectrum [56,80].

Our findings suggest the exact parent Hamiltonian for $\alpha \neq 1$ should involve more complicated U(1)-invariant interactions, potentially with larger support. We checked the hypothesis of Shastry (Ref. [54]) of considering long-range XXZ chains with square secant couplings. Up to the considered system size there is a slow trend toward larger relative entropy, thus suggesting the ansatz is likely to be insufficient. Nevertheless, finite-size results are of value for

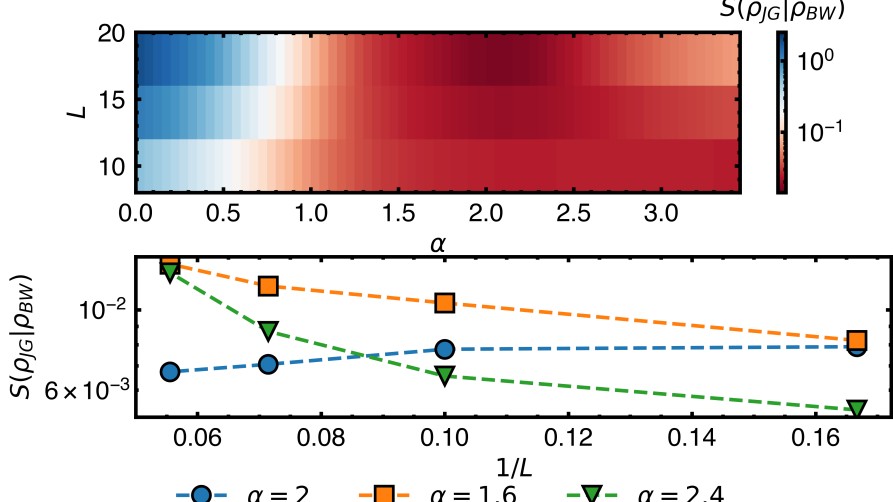

Figure 13: Relative entropy between converged BW density matrix and the JG one for the long range model Eq. (43) for $L = 8, 12, 16, 20$. The results show a decreasing relative entropy for $\alpha = 2$, which suggests the algorithm is approaching thermodynamic convergence. Instead, even points close to this Haldane-Shastry point exhibits increasing entropy, and certifying only an approximate reconstruction.

Hamiltonian engineering and quantum simulations. Indeed, the BWA method provides inherently finite-size optimization and control on the basis chosen and on the quality of the outputs. In particular one can choose experimentally suitable operators in the basis, such as two-body operators. The fact that our technique is easily adaptable to include fully-long-ranged interactions may also be used in a different manner, that is, to certify and validate quantum simulators aimed at finding ground states of spin models including slowly-decaying power-law interactions, which are realized in both trapped ions [27] and Rydberg atom experiments [25, 108].

It is of primary interest to apply similar techniques and considerations to two dimensional wave functions, such as the Laughlin wave functions. In fact, being the only computational demanding part of the algorithm the calculation of the ground state and the Bisognano-Wichmann expectation values, in principle one can tackle also higher dimensions by using Monte Carlo techniques. From the quantum engineering viewpoint, another intriguing perspective is to search for Liouvillians that have Jastrow-Gutzwiller wave functions as unique steady states [109, 110]. In particular, dissipation may considerably soften the requirement for long-range couplings thanks to correlations induced by the bath.

# Acknowledgements

We acknowledge useful discussions with G. Giudici, A. Lerose, N. Lindner and T. Mendes-Santos.

**Funding information**   This work is partly supported by the ERC under grant number 758329 (AGEnTh), and has received funding from the European Union's Horizon 2020 research and innovation programe under grant agreement No 817482.

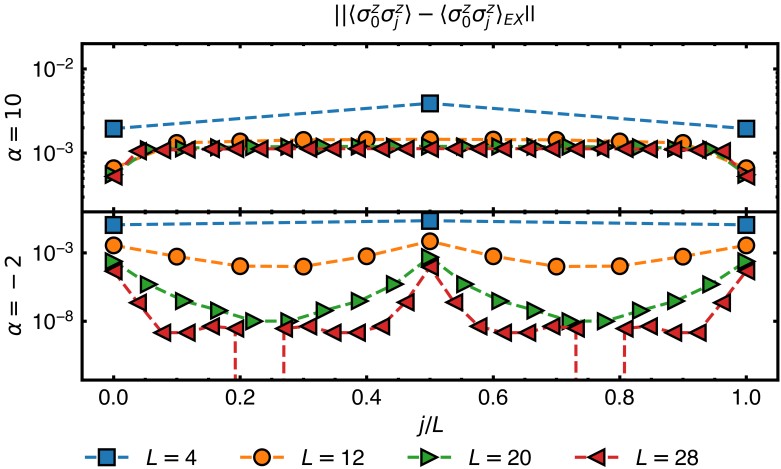

Figure 14: Difference between numerical correlation functions computed on the JG states and the analytic formulae Eq. (49). The different system sizes show a scaling to zero, confirming the correctness of the GHZ limit.

## A  Correlation functions and parent Hamiltonian for the GHZ regimes

We argued that the JG states at $\alpha < 0$ and $\alpha \gg 1$ corresponds to ferromagnetic and antiferromagnetic cat states. A first check is given by means of the participation spectrum and of the entanglement entropy (see Fig 2 in Section 2). Given the simple form of these GHZ states Eq. (9), we can compute their analytic correlation functions:

$$\langle \sigma_0 \sigma_j \rangle_c^{\alpha<0} = 2\left|1 - 2\frac{j}{L}\right| - 1, \quad \langle \sigma_0 \sigma_j \rangle_c^{\alpha \gg 1} = (-1)^j. \tag{49}$$

In Fig. 14 we check the agreement between the above equations and the numeric correlation functions computed on the exact JG states. Our results suggest the state is in a symmetry broken phase [23]. Intuitively, we can guess classical parent Hamiltonians having these states as the ground state. For example, a ferro/antiferro-magnetic Ising model with the constraint of having zero magnetization. In practice, one can represent these states as MPS and use well-known results [11, 23] to reconstruct local parent Hamiltonians.

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
