# Peer review of "Parent Hamiltonian Reconstruction of Jastrow-Gutzwiller Wavefunctions"

_SciPost Physics, doi:SciPost Phys. 8, 042 (2020)_

## Round 2 · Referee Report · Anonymous (Referee 4) · 2019-11-6

Strengths

1- The article is well written and well organised.

2- The numerical analysis is trustable.

3- The argument is timely and interesting.

Weaknesses

1- Important references on Jastrow-Guttzwiller wave functions are missing.

2- A few more calculations would improve the quality of the work.

Report

The work by Turkeshi and Dalmonte present a numerical analysis to obtain (almost) parent Hamiltonians for Jastrow-Gutzwiller wave functions in 1D.

I think that the paper is nice and should be published with a few corrections.

1) Since spin-liquids are mentioned in the introduction, I think that the recent works on frustrated Heisenberg models that employed Gutzwiller-projected states must be cited. among them:

*) For the j1-j2 model on the square lattice: Phys. Rev. B 88, 060402(R) (2013)
*) For the j1-j2 model on the triangular lattice: Phys. Rev. B 93, 144411 (2016)
*) For the Heisenberg and j1-j2 models on the Kagame lattice: Phys. Rev. B 84, 020407(R) (2011) and Phys. Rev. B 91, 020402(R) (2015)
*) The mapping to a classical coulomb gas was first suggested in Phys. Rev. B 73, 245116 (2006)

2) It would be nice to have a more systematic size scaling of S_{vN} shown in Fig.1 for only two values of L. Indeed, for large positive values of \alpha it saturates to log(2), while for large negative values it diverges. Is it possible to understand better the intermediate regime?

3) I am confused about Fig.3: I expected a finite correlation length for large values of \alpha: why is instead diverging?

4) I am also confused about Table 1: since h=0, there are only two free parameters in the (almost) parent Hamiltonian, namely \Delta_1 and J_1, so one of them can be fixed to 1.

5) At the beginning it is written that the Jastrow-Gutzwiller wave functions are good approximations of the (almost) parent Hamiltonian. For that, It would be useful to compare the exact ground-state energy of a given Hamiltonian (with NN or NN+NNN couplings) to the variational energy of the Jastrow-Gutzwiller state. An insightful plot would be to report the accuracy (E_ex-E_var)/E_ex as a function of the number of distances included in the (almost) parent Hamiltonian, for the optimal couplings on each case.

In summary, I think that the paper is nice and should be published after some modifications.

---

## Round 2 · Referee Report · Anonymous (Referee 3) · 2019-12-6

Report

In this work, Turkeshi and Dalmonte address the very interesting question of parent Hamiltonians for a class of Jastrow-Gutzwiller wave functions. After giving a very pedagogical introduction to JG wave functions, the authors use a Bisognano-Wichmann ansatz to reconstruct parent models via their entanglement Hamiltonians.

Over all this study is quite original (to my knowledge) and clearly deserves publications after minor revisions and perhaps some additional discussions I suggest.

Section 2.
JG functions are discussed for any values of the parameter \alpha. Perhaps the authors could say how novel their results, in particular about the extension of the critical regime. What is the status/nature of the transition claimed at \alpha_c=4.38 (no finite size scaling is provided, while the crossing in Fig. 2 Left appears at slightly larger \alpha_c)?
The interpretation of the weights Eq. (3) in terms of pseudo energies is quite interesting. As discussed in the context of participation entropies and spectroscopies (see papers by Stephan et al, Luitz, Alet... ), the logarithmic interaction has already been observed and discussed for critical XXZ ground-states (see Eq. 5). Could you please comment on this?
Regarding the correlation functions, an additional plot in log-log scale of Fig. 3 would be very helpful to better understand the scaling.

Section 3.
The way the distance between two reduced density matrices (\rho and \sigma, see Eq 33) is estimated deals with the relative entropy. Since I am more familiar with Kullback-Leibler divergences to compare how two distributions are close, I am curious to see the difference between these two estimates. I understand they are very close, but not exactly the same.

Section 4.
The framework for the Hamiltonian reconstruction, starting with a family of JG functions is based on a strong assumption at the very beginning, which takes from granted that the reduced density matrix takes the BW form Eqs. (21,22). This seems to give reliable results only for a small range of parameters, and obviously fail when Lorentz invariance is broken at \alpha\le 0. Here comes my question: why the authors did not try to compare their reconstruction approach with the simplest one based directly on the Hamiltonian itself by coupling the covariance matrix over a family of target hamiltonian (See Refs. 38,39,85). This methods appear easier since it simply requires to but a small matrix with computed correlation functions. I am curious to see the difference of results they would obtain. I would expect at least a comment on this aspect.

Requested changes

1- Comment on the critical alpha=4.38
2- Comment on the links with participation spectroscopy regarding the weights Eq. (5)
3- Add a log-log plot of Fig. 3
4- Comment on the alternative KL divergences measures as compared to relative entropies
5- Comment on the alternative reconstruction method based on the covariance matrix.
6- Small typo page 18 |jac) -> |jas)

---

## Round 3 · Referee Report · Anonymous · 2020-2-15

Strengths

The paper report calculations on simple Jastrow wave functions that have not been considered before.

Report

The authors replied to my questions in a satisfactory way. I think that now the paper may be published.

---

## Round 3 · Referee Report · Anonymous · 2020-2-24

Strengths

Pedagogical and thorough study of an interesting family of 1D wave functions

Report

In this new version, Turkeshi and Dalmonte have addressed several new aspects of this interesting problem. They certainly raised many new interesting effects, which deserve further studies.

I believe this manuscript is now ready for publication.

---

## Round 3 · Author Response

Dear Editor,

thank you for handling our submission.

We are pleased to thank the referees, whose insightful comments and observations helped us further improve and clarify our work. In this resubmission, we believe to have addressed all the point raised by the Referees. We append below a detailed response to the reports and the list of changes.

Yours sincerely,

Xhek Turkeshi and Marcello Dalmonte

Reply to referee report I:

We thank the referee for their thoughtful reading on the manuscript. Here we reply to the corresponding points of their report, specifying the changes included in the resubmitted manuscript:

1) We updated the bibliography with other relevant works we have missed, including the ones suggested by the referee.

2) We included additional system sizes in Fig.2 (Fig.1 of previous submission) of the paper. We also performed a better system size analysis, included in Sec. (2.3) of the new manuscript. Since the scaling is very slow, we cannot give a thermodynamic answer on the intermediate regime, which seems representative of a Luttinger liquid phase.

3) In our work we use the definition of the correlation length in terms of the real space correlation functions (eq.(21-22) of the new manuscript). These formulae are meaningful only when the cluster decomposition principle holds. However, due to the functional form of the JG wave functions, in the symmetry broken phases this state would result in a coherent superposition of states in the different symmetry broken sectors. In fact these are the GHZ states, which are a remarkable exception of the cluster decomposition. All the data presented are in the critical regime, where this principle holds (and the correlation length is well defined). To confirm this fact, we added a new subplot of the correlation functions in LogLog scale. The seemingly algebraic decay guarantees the principle is holding. We added a discussion to further clarify this point in Sec.(2.4). Finally, for presentation convenience we decided to present in the new manuscript only L multiples of 4, with system sizes up to L=36.

4) As the referee observe, the optimization with only nearest-neighbors indeed requires only one free parameter. Nevertheless, to benchmark the algorithm, we decided to present the results of an unconstrained optimization. From the table one can see that the symmetries are always respected and the relative ratios of the converged couplings are consistent with those expected in the exact cases. In the table caption we further comment on this point. We apologize for not stating this before, as this was clearly confusing.

5) Motivated by the insightful comment of the Referee, we computed the relative error between the exact ground state energy of the reconstructed Hamiltonian and its variational energy on the Jastrow-Gutzwiller state. The results are presented in a new paragraph of section 4.3. Below we summarize the included discussion, and refer to the new version of the paper for further details.

All the cases considered in our studies lie within 1\% of relative error in the energy landscape. In addition, we also present data of the relative error as a function on nearest-neighbors considered in the reconstruction. Interestingly, this quantity seems to increase with larger basis considered, although the two states are closer and closer (see Fig.5 and Fig.6). This is reminiscent of what happens for the correlation functions (Section 4.3, paragraph: Correlation functions). In the same perspective, at present we cannot fully understand and characterize such counterintuitive behavior. As we mention, this may be due to the algorithm forcing the optimization on a finite size landscape and creating frustration effects. For example, this is probably what happens in the case $\alpha=2$, which should converge to the Haldane-Shastry prefactors. Another possibility is that new operator content is needed, and the chosen basis cannot grasp the thermodynamic properties of the systems. Further investigations on this problem are left for future studies.

Reply to referee report II:

We thank the referee for their interesting comments. We address the requested changes below.

1) We added a reliable system size scaling in Sec.(2.3). We construct a mesh of the critical exponent and the critical value of alpha e consider the optimal fit over different polynomial degree and system sizes. The optimal fit is chosen using the least-square test. Values and error of the different estimates are the average and standard deviation over the set of optimal fits obtained with different L-ranges and deg-ranges. In particular, the new estimated critical alpha is around 4.3(1). 
Nonetheless, we also argued that this scaling is very slow, since is an algebraic function of Log(L). As such, further system sizes are needed to fully characterize the transition point.

3) We added a new subsection about the participation spectrum in Sec.(2.2). In particular we discuss the relation to the logarithmic potential observed in the domain walls sector of the XXZ model. We also discuss their relationship to the recently proposed conjecture that JG states are representatives of a Luttinger liquid phase. At the time of writing this paper we were not aware of the work of Ref.[64-65]. In particular in the latter the authors use the Resta polarization to numerically estimate a critical point between a Luttinger phase to a Neel ordered phase at alpha_c=4. Our simulations for similar system sizes in the entanglement entropy seems to give another value of critical transition point. We believe that further studies are needed to resolve the nature of the variational JG wave functions.

3) We have split Fig.5 ( Fig.3 of the previous version) in two LogLog subplots presenting respectively the connected correlation functions and the inverse correlation length. For notational convenience we present only L multiples of 4, with system sizes up to L=36.

4) We added a comment in Sec.(3.3) on other choices of estimators (including the KL divergence). In particular we remark this is not unique and we have chosen the relative entropy for implementation convenience. In fact the latter is both convex and has a simple derivative (used in the gradient descent). Furthermore, we mention that at present the relationship between the KL divergence and the quantum relative entropy has not been studied yet and deserve independently to be investigated.

5) As discussed in the work by Chertkov and Clark, methods based on the quantum covariance matrix, do not return the parent Hamiltonian, but rather, an Hamiltonian whose input vector is a generic eigenstate, not necessarily the ground state. In order to verify that the given state is the ground state, a separate procedure is required. As such, we feel that a direct comparison is not possible, as the methods are addressing quite different questions (out method targets the ground state by construction). 
Following the Referee’s remarks, we have expanded our discussion on this issue in the beginning of Sec.(3).

---

## Round 3 · List of Changes

* * *
List of changes:
* * *
-Corrected various typos
-Added Sec(2.2) discussing the participation spectrum (new figure Fig.1) (comments requested by Referee II)
-Added a discussion on the finite size scaling in sec 2.3 (new figure Fig.4) (to respond to both referees)
-Removed the appendix on finite size scaling
-Added new subfigure in fig.5 (old fig.3), both in Loglog scale (to respond to both the referees)
-Added a discussion in sec (2.4) about the correlation length (requested by Referee I)
-Added a discussion in the beginning of sec (3) to comment on the quantum covariance matrix approach in relationship to our work (requested by Referee II)
-Added a comment on other estimators in sec (3.3), especially the KL divergence (requested by Referee II)
-Added a clarification in the label of Table 1 (requested by Referee I)
-Added a new paragraph in sec (4.3) discussing the relative error between the variational energy of the JG wave functions and the ground state energy of the parent Hamiltonian (requested by Referee I)
-Updated bibliography

---

## Editorial Decision

published